# Enhancing Voice Cloning Quality through Data Selection and Alignment-Based Metrics †

**Ander González-Docasal** [1,2,*,‡] and **Aitor Álvarez** [1,‡]

1   Fundación Vicomtech, Basque Research and Technology Alliance (BRTA), 20009 Donostia-San Sebastián, Spain; aalvarez@vicomtech.org
2   Department of Electronics, Engineering and Communications, University of Zaragoza, 50009 Zaragoza, Spain
*   Correspondence: agonzalezd@vicomtech.org
†   This paper is an extended version of our paper published in IberSpeech 2022, Granada, Spain.
‡   These authors contributed equally to this work.

**Abstract:** Voice cloning, an emerging field in the speech-processing area, aims to generate synthetic utterances that closely resemble the voices of specific individuals. In this study, we investigated the impact of various techniques on improving the quality of voice cloning, specifically focusing on a low-quality dataset. To contrast our findings, we also used two high-quality corpora for comparative analysis. We conducted exhaustive evaluations of the quality of the gathered corpora in order to select the most-suitable data for the training of a voice-cloning system. Following these measurements, we conducted a series of ablations by removing audio files with a lower signal-to-noise ratio and higher variability in utterance speed from the corpora in order to decrease their heterogeneity. Furthermore, we introduced a novel algorithm that calculates the fraction of aligned input characters by exploiting the attention matrix of the Tacotron 2 text-to-speech system. This algorithm provides a valuable metric for evaluating the alignment quality during the voice-cloning process. We present the results of our experiments, demonstrating that the performed ablations significantly increased the quality of synthesised audio for the challenging low-quality corpus. Notably, our findings indicated that models trained on a 3 h corpus from a pre-trained model exhibit comparable audio quality to models trained from scratch using significantly larger amounts of data.

**Keywords:** voice cloning; speech synthesis; speech quality evaluation





## 1. Introduction

Voice cloning, a rapidly evolving research area, has gained significant attention in recent years. Its main objective is to produce synthetic utterances that closely resemble those of a specific speaker, referred to as the cloned speaker. This technique holds significant potential in various domains, particularly in the media industry. Applications include long-form reading of textual content, such as emails and web pages, audio book narration, voice-overs, dubbing, and more [1]. The rising demand for voice cloning can be attributed to the significant advancements in deep learning techniques, which have led to notable improvements in the quality of these systems [2].

However, as we evidenced in our previous study [3], ensuring the quality of the input data is highly important to obtain accurate results when employing voice-cloning techniques based on deep learning algorithms. It is essential that the input audio files for a specific speaker possess optimal acoustic conditions, as the cloning algorithm will replicate the training material, including any noise or audio artefacts present in the signals. This encompasses the need for minimal audio compression and optimal sampling and bit rates. Furthermore, the heterogeneity of linguistic characteristics is closely associated with the quality of a voice-cloning corpus. High variability in features such as prosody, pitch, pause duration, or rhythm can have a detrimental impact on the training of a voice-cloning system.

Addressing this issue may need a larger volume of data and/or manual annotations to ensure the satisfactory quality of the final cloned output.

Another critical challenge in the development of voice-cloning systems pertains to the measurement of the synthetic voice's quality [4]. The Mean Opinion Score (MOS) constitutes the most-precise metric for evaluating voice quality [5], which needs human evaluators to manually listen to the synthesised audio and rate it on a scale ranging from 1 to 5. Nevertheless, due to its nature, this approach primarily serves as a means of evaluating the quality of the final audio and is not suitable for assessing the quality of during-train checkpoints. In addition, the utilisation of a subjective metric that is dependent on the perception of human evaluators to assess cloned voices may produce variable results based on the unique circumstances of each evaluator [6]. Furthermore, the recent trend of MOSs in voice-cloning systems approaching levels near those of real human speech reveals the limitations of this metric in terms of comparing different models [7].

As a means of overcoming these two main challenges associated with evaluating voice-cloning systems—the impact of the input data and the lack of objective measurements—we propose an alternative evaluation framework that strives to achieve the following objectives: (1) calculating the variability of a given voice-cloning dataset in order to filter unwanted training material, (2) using objective metrics that measure the quality of the generated signals, and (3) conducting these measurements during the training process in order to monitor model improvement or the lack thereof.

This work is an extension of the authors' previous study [3], where we examined two real use cases of voice-cloning system construction under highly challenging data conditions, characterised by a small quantity of highly variable and low-quality data. We analysed one successful and one unsuccessful case using objective metrics, without conducting exhaustive data curation. In this work, we expanded upon the previous study by applying data selection techniques to the same successful case presented in the previous paper. Additionally, we introduced two new cases using two distinct high-fidelity voice-cloning datasets publicly available in the community, one in English and the other in Spanish, and with varying corpus sizes. Furthermore, we used two objective evaluation metrics based on MOS estimation models and introduced a novel algorithm for calculating the sentence alignment of the synthesised audio at the character level. Through this, we propose a methodology that concludes that data selection improves, or at the very least equates to, the quality of the voice-cloning system used in this evaluation. Moreover, we propose an iterative evaluation of models during training using the aforementioned objective metrics, providing support for the construction of these systems before subjectively evaluating the final model.

To begin, the public tool Montreal Forced Aligner (MFA) (https://montreal-forced-aligner.readthedocs.io accessed on 4 July 2023) was utilised to perform forced alignment on each corpus. Subsequently, after excluding non-aligned audio files, the alignments were used to calculate various quality metrics on the datasets, including the Signal-to-Noise Ratio (SNR) and utterance speed. These measurements allowed eliminating audio files that introduced higher variability. Various sets of models were trained both with and without these collections of more-irregular data.

As our voice-cloning framework, a Text-To-Speech (TTS) approach using the neural acoustic model Tacotron-2 [7] was adopted, which is a well-established model in the speech synthesis community. We trained various models using the aforementioned audio datasets, both with the complete versions and after excluding the subsets that introduced higher variability. The spectrograms generated by the Tacotron-2 model were transformed to waveform using the publicly available Universal model of the vocoder HiFi-GAN [8].

To gauge the efficacy of the proposed voice-cloning system, various quality evaluation metrics were employed. For evaluating the quality of the generated audio without a reference signal, two distinct MOS estimators were employed: NISQA [9] and MOSnet [4]. Additionally, a novel algorithm was introduced in order to determine the percentage of aligned characters in the attention matrix of the model as a final metric, which excluded

possibly unaligned audio files. The aforementioned measurements were conducted on the distinct checkpoints obtained during the training process of these datasets to monitor the models' progress over time.

As general conclusions, we can highlight that the difficulty of the dataset corresponding to the successful case presented in our previous work [3] was due in part to the quality of the data, as well as to the lack of enough training data. However, we managed to prove that reducing the variability of this corpus by excluding specific subsets from the training partition improved the quality of the generated audio. In the case of the contrasting High-Quality (HQ) datasets, the addition of more variable data did not necessarily imply an improvement of the synthesised audio when using the same training configuration as the difficult corpus, although the models trained from scratch using a higher amount of data were more notably influenced by these changes.

The remainder of this paper is structured as follows: Section 2 explores the related work in the research field of voice cloning. Section 3 includes the analysis of the two main corpora utilised in this work, whilst Section 4 describes the voice-cloning system and its training framework. Section 5 explains the different training scenarios and metrics used for the evaluation presented in Section 6. Finally, Section 7 draws the main conclusions and the lines of future work.

## 2. Related Work

The field of voice cloning has witnessed significant advancements in recent years, mainly driven by the remarkable progress in deep learning techniques. Numerous studies have explored various approaches and methodologies to tackle the challenges associated with this research field.

In terms of applications of voice cloning, a prominent one is observed within the scope of deep faking [10,11]. An illustrative instance is the interactive artwork released by the Salvador Dalí Museum, featuring a deep fake representation of the renowned artist [12]. In terms of audio only, the AhoMyTTS project focuses on generating a collection of synthetic voices to aid individuals who are orally disabled or have lost their own voices [13]. Similarly, the Speech-to-Speech Parrotron [14] model serves the purpose of normalising atypical speech by converting it to the voice of a canonical speaker without speech disorders, thereby enhancing its intelligibility. Another successful voice cloning endeavour was demonstrated in the Euphonia Project [15], wherein the voice of a former American football player diagnosed with amyotrophic lateral sclerosis was recovered through the utilisation of a database containing his recordings [16]. Finally, the voice of former Spanish dictator Francisco Franco was cloned for the emission of the XRey podcast [17], winner of the Best Podcast Ondas Award in the National Radio category, needed for the synthesis of a letter and an interview in a process explained in [3] by the authors.

This research field can be broadly categorised into two main branches: voice conversion and speech synthesis or TTS. Voice conversion aims to transform the characteristics of a source speaker's voice into those of a target speaker, while preserving the linguistic content and speech quality of the original audio. In the past few years, the efficacy of deep learning techniques in voice conversion has been well-established. Various architectures including autoencoders have gained popularity for this purpose, such as variational autoencoders [18] and bottleneck-based autoencoders [19,20]. Researchers have also explored the application of Generative Adversarial Networks (GANs) in this specific task [21–23]. Additionally, deep feature extractors have been leveraged to achieve successful outcomes. For instance, van Niekerk et al. [24] employed an approach based on HuBERT [25] for many-to-one voice conversion.

On the other hand, deep learning techniques have emerged as the leading approach in the field of text-to-speech systems. WaveNet [2], which employs dilated Convolutional Neural Networks (CNNs) to directly generate waveforms, marked a significant milestone in this domain. Since then, numerous neural architectures have been developed by the scientific community. With Tacotron-2 [7], an attention-based system combined with a set of

Long Short-Term Memory layers, many acoustic models that generate a Mel frequency spectrogram from linguistic input have been developed. Some of these architectures employ a phoneme length predictor instead of an attention matrix [26–28]. Additionally, sequence-to-sequence modelling has gained traction in this research field. Mehta et al. [29], for example, proposed the use of neural Hidden Markov Models (HMMs) with normalising flows as an acoustic model. In the context of generating a speech signal from Mel spectrograms, vocoders based on GANs [8,30–32] have gained popularity due to their efficient inference speed, lightweight networks, and ability to produce high-quality waveforms. Furthermore, end-to-end models such as VITS [33] or YourTTS [34] have been developed, enabling the direct generation of audio signals from linguistic input without the need for an additional vocoder model. Finally, important advances have been made in terms of zero-shot TTS systems that feature voice conversion with the use of a decoder-only architecture. As an example, the system VALL-E is capable of cloning a voice with only 3 s of the target speaker [35].

With regard to the corpora, numerous many-to-one voice cloning corpora are available in the community, primarily designed for TTS approaches. These corpora prioritise the measurement of dataset quality across various dimensions. Regarding signal quality, the SNR holds significant importance, both during the content filtering [36,37] and data recording stages [38–41]. Linguistic considerations also come into play, with some researchers emphasising the need for balanced phonemic or supraphonemic units within the dataset [38,39,41,42]. Additionally, text preprocessing techniques are employed to ensure accurate alignment with the uttered speech and to reduce variability in pronunciations [36,39–44]. Lastly, the quantity of audio data generated by each speaker is a critical aspect in corpus creation, particularly in datasets with a low number of speakers [36,38–44].

Finally, to evaluate the quality of voice-cloning systems, various approaches have been devised within the research community. Objective metrics that compare a degraded signal to a reference signal, such as PESQ, PEAQ, or POLQA [45], are widely available for assessing audio fidelity in scenarios involving, e.g., signal transfer or speech enhancement. However, these metrics exhibit limitations when applied to voice-cloning systems, as the degraded signal may not be directly related to the original one. Consequently, an emerging trend focuses on estimating the MOS using the cloned signal alone. MOSnet [4,5] and NISQA [9,46] are examples of systems that employ deep learning models for MOS estimation in this context. In line with this trend, the VoiceMOS Challenge [47] was introduced to address the specific issue of MOS estimation in voice-cloning systems. The challenge served as a platform for the development of this type of system, fostering advancements in the field and facilitating the resolution of this particular challenge.

Our study focused on evaluating the quality of a particularly complicated voice-cloning dataset featured in [3] with the aim of identifying and removing audio files that contribute to heterogeneity within the overall data. The impact of these reductions was assessed within a TTS framework utilising a Tacotron-2 model as the acoustic model, paired with a HiFi-GAN-based vocoder. Furthermore, the quality of the resulting audio was evaluated using MOSnet and NISQA as the MOS estimator systems. Additionally, two distinct open voice cloning corpora in English and Spanish were also processed for contrasting purposes.

## 3. Audio Datasets

The training and evaluation framework for voice-cloning systems was put to test using the database *XRey*, primarily described in our previous work [3]. It mainly consists of utterances of Spanish dictator Francisco Franco, whose voice was cloned for the podcast XRey [17], winner of an Ondas award in 2020, which recognises the best Spanish professionals in the fields of the radio, television, cinema, and music industries, and it is available in Spotify with an added special track (https://open.spotify.com/episode/0Vkoa3ysS998PXkKNuh9m2 accessed on 4 July 2023) in which the generation of the cloned voice is explained by the authors in detail.

Additionally, a set of two freely available HQ datasets was chosen, each in a different language: English and Spanish. It was ensured that both corpora were publicly available and comprised a sufficient number of audio hours for training multiple Tacotron-2 TTS models on different training subsets.

### 3.1. XRey

The corpus for *XRey* is mainly composed of utterances from Christmas speeches from the years ranging from 1955 to 1969 divided into three acoustically similar groups of roughly 1 h each, on a total of 3:13 h of speech on 1075 audio files. These files were obtained from YouTube and the RTVE play (https://www.rtve.es/play/ accessed on 4 July 2023) web portal and were automatically transcribed, manually post-edited, and forced-aligned using the Kaldi toolkit [48] as explained in our previous work [3]. The final audios were upsampled from 16 kHz 16 bit WAV PCM to 22,050 Hz. More-detailed information about the features of this dataset can be found in Section 3.4.

Even though the speeches that compose this corpus are publicly available, the postprocessing and manual annotation make it a private dataset, which, due to the considerations of this particular personality, will not be released to the public.

### 3.2. Hi-Fi TTS

The corpus chosen for the English language was the dataset *Hi-Fi TTS* [36]. It is composed of 10 different speakers, 6 female and 4 male, where each speaker has at least 17 h of speech obtained from recordings from audio books from LibriVox. Each audio file in the dataset is categorised as belonging to the subsets *clean* or *other* according to the values of the SNR. From this multi-speaker corpus, the female speaker with ID 92 (https://librivox.org/reader/92 accessed on 4 July 2023), with a total of 27:18 h in 35,296 audio files of *clean* speech no longer than 20 s each, was chosen as the English voice for this work. These audio files were originally found in mono MP3 format sampled to 44,100 Hz, although they were converted to 22,050 Hz and 16 bit WAV PCM. More-detailed information about the features of this dataset can be found in Section 3.4.

### 3.3. Tux

The audio dataset chosen for the Spanish language is composed of around 100 h of audio books recorded by a single speaker from LibriVox, the user Tux (https://librivox.org/reader/3946 accessed on 4 July 2023). It is divided in two different subsets: *valid* and *other*. The *valid* subset was reviewed with an automatic speech recognition system based on DeepSpeech, composed of the audio files whose automatic transcriptions and original text match, constituting a total of 53:47 h of speech in 52,408 audio files no longer than 24 s each. The audios were found in mono and sampled to 22,050 Hz 16 bit WAV PCM. This corpus was processed and released to the public by Github user `carlfm01` (https://github.com/carlfm01/my-speech-datasets accessed on 4 July 2023). More-detailed information about the features of this dataset can be found in Section 3.4.

### 3.4. Data Analysis

During this stage of the pre-training corpora evaluation, the acquired data must be analysed for a posterior cleaning thereof. To this end, the first required step is the forced alignment of the audio and text files. The Montreal Forced Aligner (MFA) was chosen for this purpose since both the tool and the models are publicly available. The model `spanish_mfa` v2.0.0a [49] was employed for aligning the corpora *XRey* and *Tux*, whereas the English dataset *Hi-Fi TTS 92* was aligned using the `english_mfa` v2.0.0a [50] model. These two models implement a Gaussian Mixture Model (GMM)-HMM architecture that uses Mel Frequency Cepstral Coefficients (MFCCs) and pitch as the acoustic features and phones as the text input. The `spanish_mfa` model was trained on 1769.56 h of audio content, while the `english_mfa` model used 3686.98 h for training. The alignment configuration had a beam width of 1 and a retry beam of 2. The pipeline discarded any audio files that were

not appropriately aligned during this process, resulting in a total of 51,397 (98.07%) for *Tux* and 35,007 files (99.18%) for *Hi-Fi TTS*. This process did not discard any files from *XRey* since they had been already force-aligned in the dataset preparation stage, as described in our previous work [3].

As the next step of the evaluation pipeline, using the information gathered from the forced alignments—mainly time marks of individual phones and words—different measurements were performed on the audio datasets, more specifically: phonetic frequency, SNR, and uttering speed.

### 3.4.1. Phonetic Frequency

One of the key aspects of developing a corpus for voice-cloning applications, especially in a TTS setup, is that it should be phonetically balanced in order to contain a representative number of samples of each phonetic unit [38,41,42]. Following this approach, we measured the phonetic content of both corpora in terms of the frequency of phones and diphones.

Accounting for 26 Spanish and 38 English phones, a total of 386 diphone combinations for *XRey*, 446 for *Tux*, and 1255 for *Hi-Fi TTS 92* were found. It should be noted that not all of the $26^2 = 626$ and $38^2 = 1444$ diphone combinations are phonotactically possible in Spanish and English, respectively. The distribution of diphones in both datasets is illustrated in Figure 1.

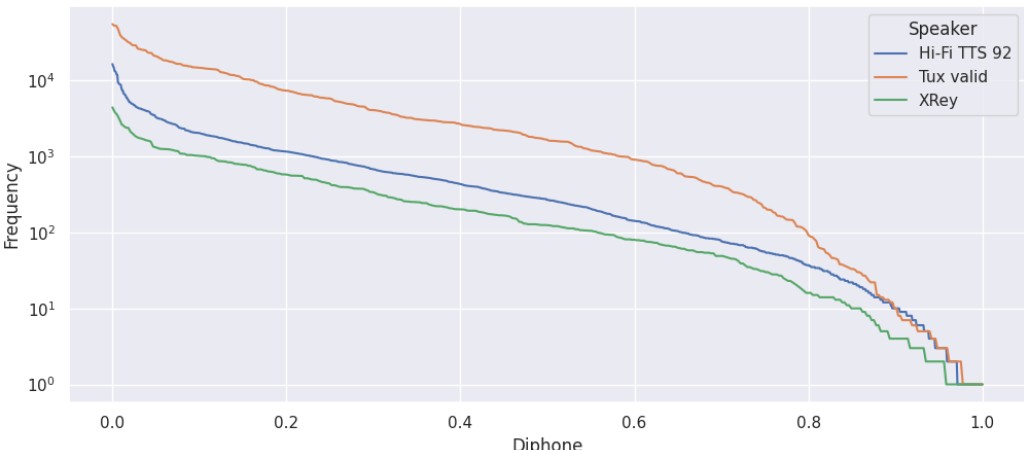

**Figure 1.** Absolute frequencies of diphones for the speaker *XRey* (green), the *valid* subset of the *Tux* corpus (orange), and the speaker 92 of *Hi-Fi TTS* (blue). The x axis is normalised to the maximum number of diphones for each speaker (386, 446, and 1255, respectively).

The obtained diphone distribution curves are similar to those computed in our previous work [3]. These findings confirm that a significantly large text corpus results in the frequencies of phones and diphones conforming to the same numerical distributions.

### 3.4.2. SNR

The next metric to be evaluated for these voice-cloning datasets is the Signal-to-Noise Ratio (SNR). For the *Hi-Fi TTS* dataset, Bakhturina et al. highlighted the importance of audio quality in terms of the SNR [36]. After estimating the bandwidth of the speech signal, they calculated the SNR by comparing the noise power in both speech and non-speech segments using a Voice Activity Detection (VAD) module. The *clean* subset was composed by audio files of a minimum SNR value of 40 dB in the 300 Hz to 4 kHz frequency band, while the files with a minimum SNR value of 32 dB fell into the *other* subset. For speaker *XRey*, however, the audio quality was relatively lower due to the recordings being conducted in the third quarter of the 20th Century. In our previous work [3], the SNR values were measured using a Waveform Amplitude Distribution Analysis (WADA) estimation algorithm.

In this study, the forced audio alignments generated in the previous step were used to obtain speech and non-speech segments required for SNR calculation, instead of relying on an external VAD module or WADA-based algorithms. Using that information, the SNR values were computed by comparing the power of the speech vs. the non-speech segments, supposing that the speech signal is the sum of the clean speech and the background noise obtained from the non-speech segment. Therefore, the SNR values obtained through this method may differ from those reported by the other works. The results can be seen in Figure 2 and Table 1.

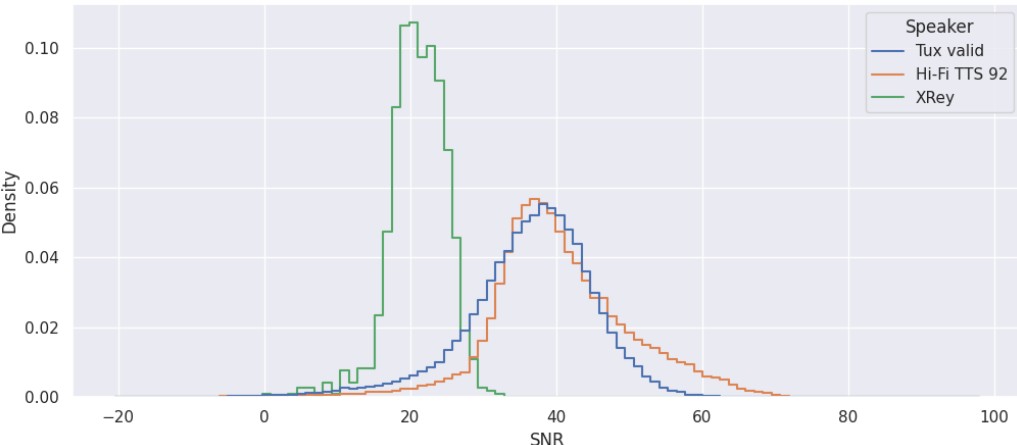

**Figure 2.** Density of the SNR values calculated by using the information of the forced alignments of *XRey* (green), subset *valid* of the *Tux* dataset (blue), and the speaker 92 from *Hi-Fi TTS* (orange).

**Table 1.** Minimum, maximum, average, median, and standard deviation values of the SNR calculated on *XRey*, on the *valid* subset of the *Tux* corpus, and on speaker 92 of the *Hi-Fi TTS* corpus.

| Speaker | Min. | Max | Mean | Median | Stdev |
|---|---|---|---|---|---|
| XRey | 0.72 | 32.81 | 21.15 | 21.27 | 3.84 |
| Tux *valid* | −20.34 | 97.90 | 36.48 | 37.36 | 8.75 |
| Hi-Fi TTS 92 | −17.97 | 73.86 | 40.57 | 39.48 | 9.62 |

Based on the data, it can be noted that the SNR values in the HQ corpora were generally high, indicating that these two datasets are suitable for voice-cloning applications with respect to the signal quality. The same cannot be said, however, for speaker *XRey*. Nevertheless, many of the audio files had an SNR value higher than 20 dB, which can be considered as sufficient quality in some speech applications.

### 3.4.3. Uttering Speed

As an effort to measure the variability of the multiple audio files that compose the different corpora, the uttering speed was also computed. Using the information obtained from the forced alignment, more specifically the duration of individual phones, the speed of each utterance $S$ can be easily obtained by dividing the number of uttered phones by the sum of the durations of each of the phones, as shown in Equation (1):

$$S = \frac{n}{\sum_i^n \mathrm{dur}(p_i)} \tag{1}$$

where $\mathrm{dur}(p_i)$ is the duration of the individual phone $p_i$ and $n$ is the total number of them. $S$ is, therefore, measured in phones per second. Notice that the duration of silences and pauses was not computed.

Using this metric, the variability of each corpus in terms of speed can be measured. The obtained results can be found in Figure 3 and in Table 2.

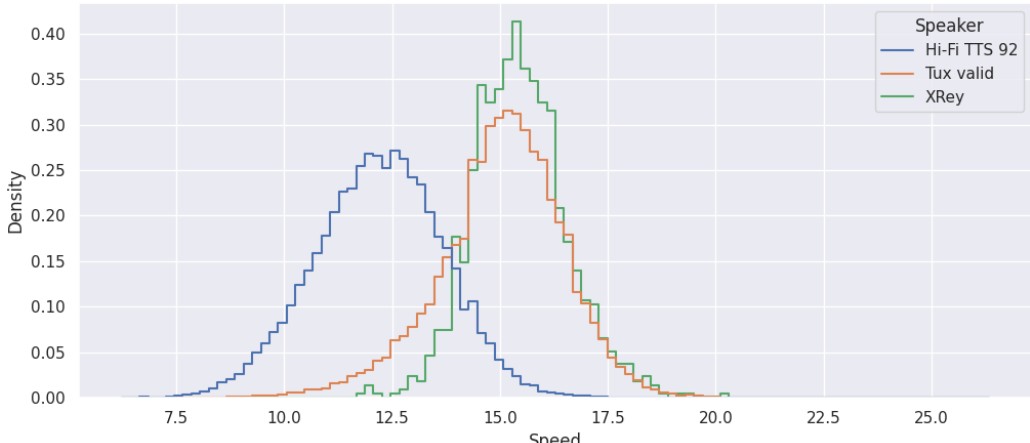

**Figure 3.** Density of utterance speed values of the speaker *XRey* (green), the subset *valid* of *Tux* (orange), and speaker 92 from *Hi-Fi TTS* (blue).

**Table 2.** Minimum, maximum, average, median, and standard deviation values of utterance speed calculated on *XRey*, on the *valid* subset of the *Tux* corpus, and on speaker 92 of the *Hi-Fi TTS* corpus.

| Speaker | Min. | Max | Mean | Median | Stdev |
|---|---|---|---|---|---|
| XRey | 11.68 | 20.11 | 15.46 | 15.42 | 1.09 |
| Tux *valid* | 6.25 | 22.22 | 12.20 | 12.23 | 1.49 |
| Hi-Fi TTS 92 | 7.46 | 26.32 | 15.03 | 15.14 | 1.46 |

Based on the data, it can be seen that the utterance speed conformed to a normal distribution in the three corpora.

## 4. Voice-Cloning System

As discussed in Section 2, the voice-cloning community has introduced various systems within the text-to-speech and voice conversion frameworks. In this study, the primary system employed for training was Tacotron-2. Although there exist newer and faster architectures for voice cloning, Tacotron-2 is a well-established system in the TTS field, acknowledged by the community. Given that the focus of this study was not to compare different architectures, but rather to evaluate the training process of a specific one and considering that this work is an extension of [3], where Tacotron-2 was utilised, it was decided to maintain the use of this architecture in the experiments conducted for this study.

Two different approaches were used for training the multiple models used in this work. For the corpus *XRey* and the subsets derived from the random partition of 3 h of the HQ datasets (refer to Section 5), a fine-tuning approach was used. Each model started using the weights of a publicly available model (https://github.com/NVIDIA/tacotron2 accessed on 4 July 2023) trained on the LJ Speech corpus [51]. These models were trained on a single GPU with a batch size of 32 and a learning rate of $10^{-4}$ for a total of 50,000 iterations.

In the case of the whole datasets, the training setup corresponded to the original Tacotron-2 recipe [7]: they were trained from scratch on a single GPU with a batch size of 64, using a learning rate of $10^{-3}$ decaying to $10^{-5}$ after 50,000 iterations, for a total of 150,000 training steps. For both approaches, the Adam Optimiser [52] was used with the following parameters: $\beta_1 = 0.9$, $\beta_2 = 0.999$, and $\varepsilon = 10^{-6}$.

The acoustic data were sampled to 22,050 Hz prior to training, and due to the characteristics of the writing systems of these two languages, the text input of the Spanish corpora was left as characters, but phonemes were used for the English dataset.

Regarding the vocoder, the Hi-Fi GAN [8] architecture was chosen, which is a GAN that employs a generator based on a feed-forward WaveNet [53] supplemented with a 12 1D

convolutional postnet. In this study, the `UNIVERSAL_V1` model was selected as the primary vocoder. This model was trained on the LibriSpeech [43], VCTK [54], and LJ Speech [51] datasets and was publicly released by the authors (https://github.com/jik876/hifi-gan accessed on 4 July 2023).

## 5. Experimental Framework

In this section, the final datasets used for training the TTS models and the quality-evaluation procedure are presented in detail.

### 5.1. Postprocessed Datasets

As was claimed before, one of the key aspects of training a voice-cloning system based on a monolingual TTS approach is the homogeneity of the data. In order to reduce the variability of the gathered corpora, three main decisions were made within the data-selection phase:

1.  Removing the sentences whose SNR value was lower than 20 dB in order to ensure a sufficient signal quality. A value of 25 dB was chosen for the HQ datasets since the quality of these corpora was notably higher (refer to Figure 2 and Table 1).
2.  Removing the sentences whose utterance speed value was inside the first or last deciles. More specifically, maintaining audio files that $14.17 < S < 16.88$ for *XRey*, $13.16 < S < 16.75$ for *Tux*, and $10.27 < S < 14.06$ for *Hi-Fi TTS 92*, where $S$ is obtained from Equation (1) (refer to Figure 3 and Table 2).
3.  Only audio files whose duration was between 1 and 10 s were used in order to reduce variability and increase the batch size during training.

Due to the lack of data for corpus *XRey* and the particularly long utterances of its speaker, the original audio files were divided based on pauses found inside sentences instead of discarding whole utterances in order to ensure that a significant part of the dataset was not lost.

With regard to the HQ datasets, in order to reproduce the difficult conditions in terms of the quantity of data found for *XRey*, a random partition of 3 h of audio was chosen from each HQ dataset, composed of 1 h of audio from the ranges of 1 to 3 s, 3 to 7 s, and 7 to 10 s. These datasets were also reduced in terms of the SNR and utterance speed. This process resulted in a total of 20 different audio collections shown in Table 3.

**Table 3.** Comparison of the number of files and hours of *XRey*, *Tux*, and speaker 92 of *Hi-Fi TTS* after removing audio files considered to increase variability. Superscripts [1], [2], and [3] correspond to the changes proposed in Section 5.1. The reader may note that the files of speaker *XRey* were divided in order to obtain shorter audio files.

| Speaker | All | | High SNR [1] | | Utt. Speed [2] | | SNR and Speed [1,2] | |
|---|---|---|---|---|---|---|---|---|
| | Files | Hours | Files | Hours | Files | Hours | Files | Hours |
| XRey | 1075 | 3:13 | 588 | 1:49 | 792 | 2:25 | 493 | 1:33 |
| Short sentences [3] | 2398 | 2:46 | 1451 | 1:35 | 1978 | 2:08 | 1249 | 1:23 |
| Tux *valid* | 52,398 | 53:46 | 44,549 | 47:21 | 38,395 | 44:16 | 33,889 | 40:33 |
| Short sentences [3] | 46,846 | 45:08 | 40,111 | 39:43 | 35,503 | 36:46 | 31,345 | 33:31 |
| 3 h partition | 3092 | 3:00 | 2649 | 2:39 | 2326 | 2:27 | 2061 | 2:15 |
| Hi-Fi TTS 92 | 35,296 | 27:18 | 31,634 | 25:02 | 25,975 | 21:40 | 23,996 | 20:16 |
| Short sentences [3] | 33,589 | 26:34 | 30,374 | 24:27 | 25,838 | 21:19 | 23,889 | 19:58 |
| 3 h partition | 3131 | 3:00 | 2835 | 2:45 | 2486 | 2:31 | 2301 | 2:21 |

The impact of these ablations was measured with multiple trainings of the voice-cloning systems as explained in Section 6.

*5.2. Quality Measurement*

The final step of the voice-cloning evaluation framework proposed in this work was to test the quality of each of the checkpoints that were generated during the training of the 20 models by using objective metrics. For that purpose, a set of various metrics was gathered. These metrics can be classified into two different categories: MOS estimators and alignment metrics.

### 5.2.1. MOS Estimators

MOS estimators typically use deep learning algorithms in order to predict the MOS score of an individual cloned signal. As an advantage, they do not require the existence of a ground truth audio. In this work, NISQA (https://github.com/gabrielmittag/NISQA accessed on 4 July 2023) [9] and MOSnet (https://github.com/lochenchou/MOSNet accessed on 4 July 2023) [4] were chosen as contrasting MOS estimator models.

### 5.2.2. Alignment Metrics

These metrics aim to obtain the number of correctly generated cloned utterances by matching the input text sequence with the resulting waveform. As an example, an estimation of the number of correct sentences can be computed by means of an automatic force aligner such as MFA, trying to match the generated audio with the corresponding input text with the lowest beam possible. However, a successful forced alignment does not ensure that a particular sentence has been correctly generated.

In this context, we propose a complementary alignment metric that takes advantage of the characteristics of the voice-cloning system Tacotron-2, its attention matrix specifically, for computing the number of input characters that have been correctly synthesised from each sentence. Given that the weights of the attention matrix for successfully generated audio should present a diagonal line, its presence can be easily checked by using a series of sliding rectangular windows:

Let $A = (a_{ij})$ be an attention matrix of dimensions $E \times D$ where $E$ is the length of the input sequence, $D$ is the length of the output spectrogram, and every $a_{ij}$ represents the attention score of the matrix. In order to check if all these elements have scored a minimum threshold value, they were clustered using sliding rectangles of a fixed size with values of the width $w$ and height $h$. The algorithm starts with the first rectangle at position $x = 0$, $y = 0$ (lower-left corner). Every element $a_{ij}$ of the attention matrix inside this rectangle is checked to be higher than a given threshold value $\theta$, which can be modified as an input parameter in order to be more or less aggressive. The rectangle is given a margin of $\frac{1}{3}w$ to the left in order to facilitate the search of aligned characters inside the region. If there exists a value $a_{ij} > \theta$ inside this rectangle, then the $i$-th input character is considered to be correctly aligned. This process is then repeated, sliding the rectangle to the position of the last correctly aligned character until:

- No correctly aligned character is found inside the region: there is no correctly synthesised character beyond this point,
- Its uppermost point exceeds $E$: the characters at the end of the sentence are aligned,
- Or its leftmost point exceeds $D$: the alignment algorithm reaches the end of the spectrogram.

The algorithmic implementation is shown in Algorithm 1, and an example of this procedure can be found in Figure 4.

In this work, all the aligned characters were calculated with the following values: $w = 150$ spectrogram windows, $h = 8$ characters, and $\theta = 0.7$, as they performed the best in our previous experiments.

---

**Algorithm 1** Character alignment algorithm using the attention matrix of Tacotron-2, given an attention matrix $A = (a_{ij})$ of dimensions $E \times D$, a rectangle of width $w$ and height $h$, and threshold value $\theta$. The algorithm is explained in detail in Section 5.2.2

---

1: **function** ALIGNED CHARACTERS$(A, w, h, \theta)$

2:     $x, y \leftarrow 0, 0$

3:     aligned_total $\leftarrow 0$

4:     **while** $y + h < E$ **and** $x + \dfrac{2}{3}w < D$ **do**

5:         # *Attention scores higher than $\theta$ inside the rectangle:*

6:         $N \leftarrow \left\{ a_{ij} \; : \; a_{ij} > \theta \; \textbf{and} \; y < i \leq y + h \; \textbf{and} \; x - \dfrac{1}{3}w < j \leq x + \dfrac{2}{3}w \right\}$

7:         # *Get the number of characters with a score higher than $\theta$ inside the rectangle:*

8:         aligned_rect $\leftarrow \left| \{ i : a_{ij} \in N \} \right|$

9:         **if** aligned_rect $= 0$ **then**

10:             **break**

11:         **end if**

12:         aligned_total $\leftarrow$ aligned_total $+$ aligned_rect

13:         # *Slide the rectangle to the position of the last aligned element:*

14:         $y \leftarrow \max\{ i : a_{ij} \in N \}$

15:         $x \leftarrow \max\{ j : a_{ij} \in N \}$

16:     **end while**

17:     **return** aligned_total

18: **end function**

---

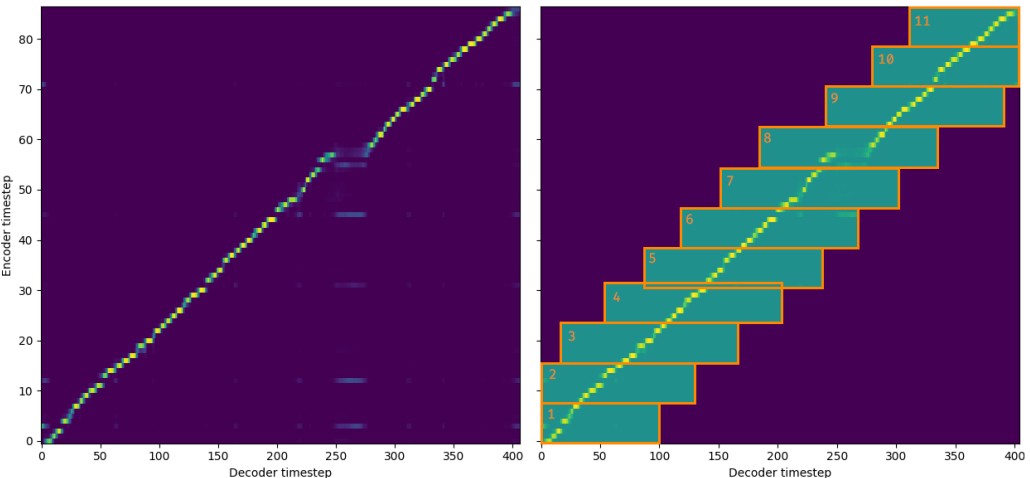

**Figure 4.** An example of an attention matrix of a decoding in Tacotron-2 (**left**) and highlighting the regions processed by the character alignment algorithm (**right**). Starting from the lower-left corner, a character $i$ is considered aligned if any value $a_{ij}$ of said row $i$ is higher than the threshold value $\theta = 0.7$ inside a rectangle of width $w = 150$ and height $h = 8$ (marked in orange as Rectangle 1). Each following step, the rectangle is slid up and right to the position of the lastly aligned character, following the diagonal, as indicated by the increasing numbers. The reader may notice that the size of the rectangle can be smaller if a region falls outside the attention matrix.

## 6. Evaluation Results and Discussion

Using the postprocessed datasets explained in Section 5.1, a total of 20 voice cloning models were trained: three different speakers—*XRey*, *Tux*, and *Hi-Fi TTS 92*—with the HQ corpora using a random 3 h partition or the whole dataset, and each of them discarding or not the audio files considered more variable in terms of the SNR and utterance speed.

As an evaluation set, a series of out-of-training sentences were gathered for each language: 207 sentences in Spanish, taken from the subset *other* of the speaker *Tux*; and 221 in English, taken from substrings of the texts corresponding to audio files whose duration was longer than 10 s of speaker *HiFi-TTS 92*. These collections of texts were synthesised each in 5000 training steps in order to compute the aforementioned evaluation metrics on these generated utterances.

### 6.1. Evaluation of XRey

This subsection presents the results of the evaluation performed on the models trained on the corpus *XRey*. For that purpose, the following metrics were used: the MOS estimation by MOSnet and NISQA, the fraction of correctly aligned sentences, and the fraction of correctly aligned characters.

The results of these metrics can be found in Figure 5.

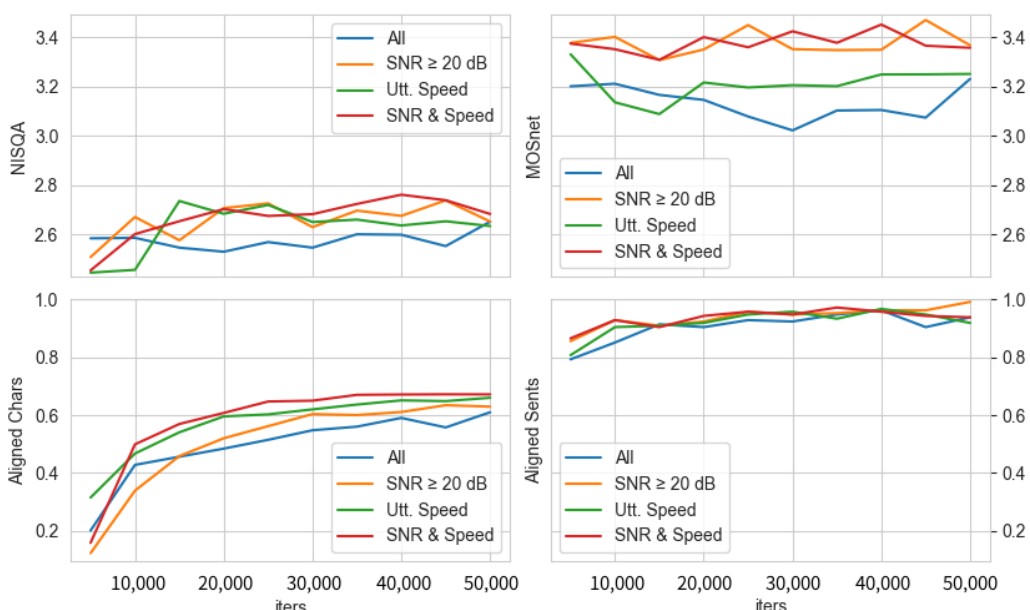

**Figure 5.** Fraction of aligned characters (**above left**) and sentences (**above right**) and the estimated MOS values obtained from NISQA (**below left**) and MOSnet (**below right**) for audio generated for each iteration for the speaker *XRey*.

The results presented in Figure 5 provide clear evidence of substantial improvement across all four measured metrics following the exclusion of audio files with high variability from the training set. It is worth mentioning that, even in the case of the most-drastic ablation, where 50% of the training data were excluded (removing the SNR and utterance speed), the resulting metrics displayed noteworthy enhancement.

Regarding the two selected MOS estimator models, it can be observed that MOSnet was significantly more generous than NISQA in this environment. Nevertheless, these MOS values indicated a considerable quality, especially for MOSnet, taking into account that the training audio files were recorded in the third quarter of the 20th Century.

In the case of the fraction of aligned sentences, it can be seen that it reached values near 100% in the early stages of training, showing that this metric was not really suitable for comparing the quality of a given voice-cloning system. The aligner was, however, performing in the way it was conceived of, that is trying to correctly align the highest possible number of sentences.

In contrast, the fraction of aligned characters did indeed show a rising trend while the training progressed. This metric, as is shown in the measurements for the other training configurations, rarely reached values near 100%, since not every input character should

have a direct effect on a particular step of the output spectrogram, particularly in the case of spaces or punctuation marks.

As the main conclusion to this case, in which low-quality audios were involved, it can be stated that excluding the audios with the higher variability of the SNR and speed levels helped not only improve the quality of the models, but also reached better MOS estimator metrics faster, even though the training data were composed of less than 3 h of audio content.

### 6.2. Evaluation of HQ Speakers Trained on 3 h

This subsection presents the results of the evaluation performed on the models trained with a random 3 h partition and their corresponding ablations as described in Section 5.1.

The metrics regarding the two MOS estimators NISQA and MOSnet on the eight different training setups corresponding to the these subsets are displayed in Figure 6.

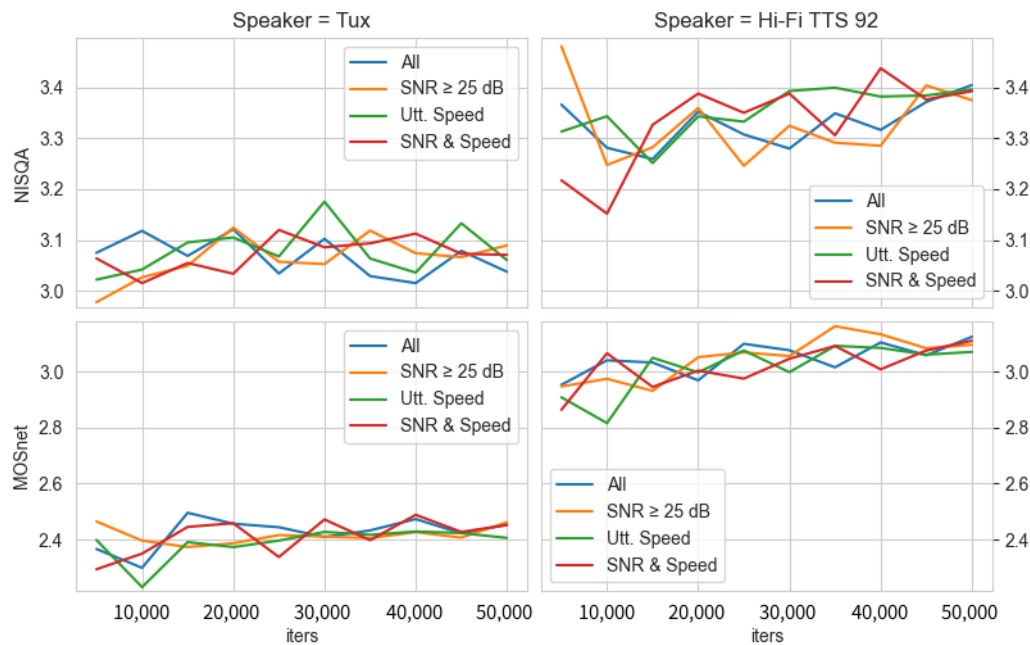

**Figure 6.** Estimated MOS values obtained from NISQA (**above**) and MOSnet (**below**) for the audio generated for each iteration for the HQ speakers *Tux* (**left**) and *Hi-Fi TTS 92* (**right**) on the random 3 h partitions.

As can be observed from the results displayed in Figure 6, there was no significant change in the estimated MOS values when comparing the models that use more data with those for which some audio files were removed. In addition, the speaker *Hi-Fi TTS 92* tended to have a higher estimated MOS value than speaker *Tux*. Both speakers had an estimated MOS value between 3 and 3.5 for NISQA and between 2.3 and 3.2 for MOSnet, concluding that NISQA is more generous in term of MOS estimation than MOSnet for these two speakers, just the opposite that happened with speaker *XRey*. In any case, it can be stated as the main conclusion of this graph that the impact of having more variable audios in the training dataset did not impact the quality of the models significantly, and therefore, adding a set of more-variable audio files to the training set did not necessarily guarantee a better final result.

In terms of sentence and character alignments, the results of the models trained on the random 3 h partitions are portrayed in Figure 7.

Similar to the speaker *XRey*, the percentage of aligned sentences using MFA approached 100% in the early stages of training, which confirmed that this metric is not the most-suitable for this particular task. Regarding character alignments, however, an improving evolution can be observed as the training progressed. Even though removing

audios with a lower SNR had a noticeable impact on the number of characters aligned for *Tux*, having less audios with more variability in terms of utterance speed did not influence the number of aligned characters significantly.

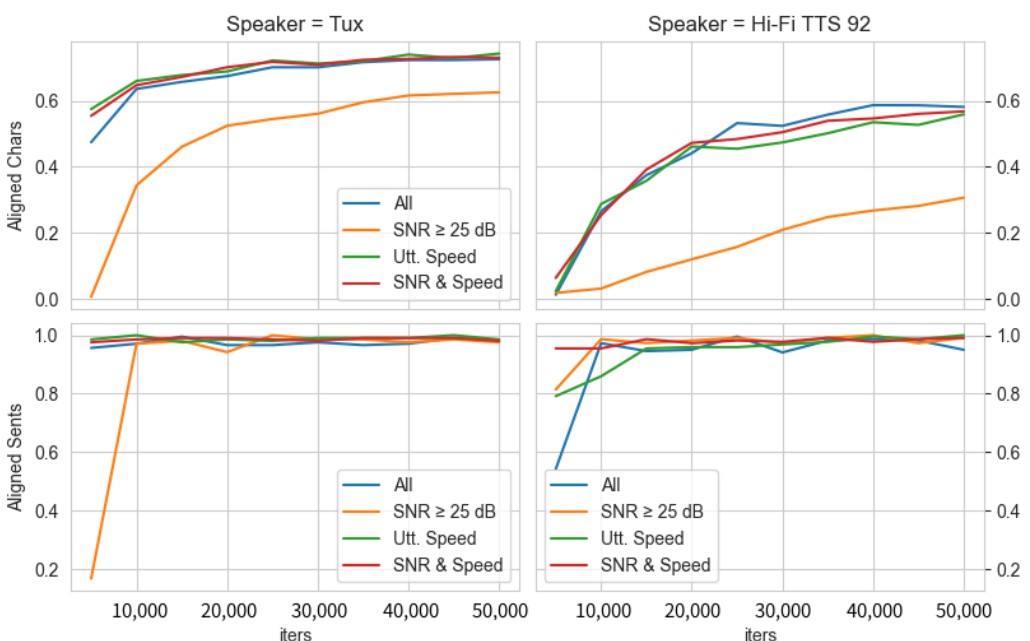

**Figure 7.** Fraction of aligned characters (**above**) and sentences (**below**) for generated audios for each iteration for the HQ speakers *Tux* (**left**) and *Hi-Fi TTS 92* (**right**) on the random 3 h partitions.

It is quite noticeable, nonetheless, that removing audio files with a lower SNR and with more variable utterance speed had a better impact on the training than removing audio files with a lower SNR only for both speakers. This could be due to the fact that these two datasets had relatively low values of noise, and therefore, the audio files whose SNR value was lower than 25 dB did not necessarily increase the variability of the data, in contrast with the utterance speed, with a similar impact as having a lower quantity of training data.

### 6.3. Evaluation of HQ Speakers Trained on the Whole Corpora

Subsequently, the evaluation results derived from the eight models that were trained using both the complete corpora and their respective ablations will be presented, as detailed in Section 5.1.

The metrics regarding the two MOS estimators NISQA and MOSnet on the eight different trainings corresponding to the corpora derived from the whole datasets are displayed in Figure 8.

One of the key findings derived from the MOS estimations obtained through NISQA and MOSnet is that the trends and values observed in the advanced stages of training were remarkably similar between models trained with random 3 h partitions and those trained with the entire corpus. Notably, Speaker *Tux* achieved values close to 3.0 for NISQA and 2.5 for MOSnet, while *Hi-Fi TTS 92* attained higher values of 3.3 for NISQA and 3.1 for MOSnet. These experiments suggested that similar quality can be achieved when training a 3 h corpus from a pre-trained model compared to a more-voluminous corpus trained from scratch, at least based on these MOS estimators.

Finally, the information regarding the fraction of aligned characters and sentences is presented in Figure 9.

Similar to the previous examples, once the fraction of aligned sentences approached the maximum value of one in the later stages of training, this particular metric had no substantial information regarding the quality of the synthesised utterances.

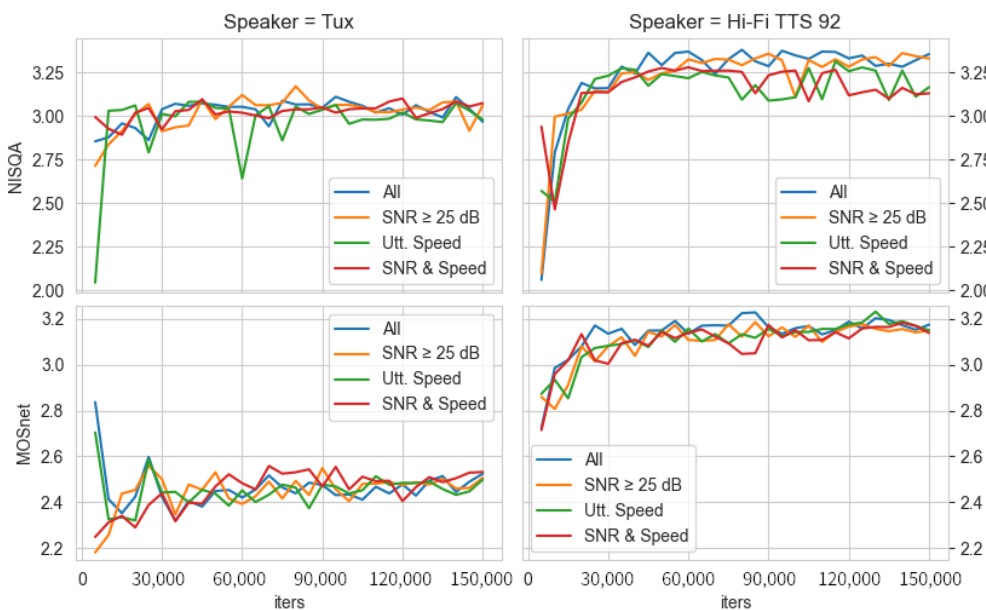

**Figure 8.** Estimated MOS values obtained from NISQA (**above**) and MOSnet (**below**) for the audio generated for each iteration for the HQ speakers *Tux* (**left**) and *Hi-Fi TTS 92* (**right**) trained from scratch on the whole corpora.

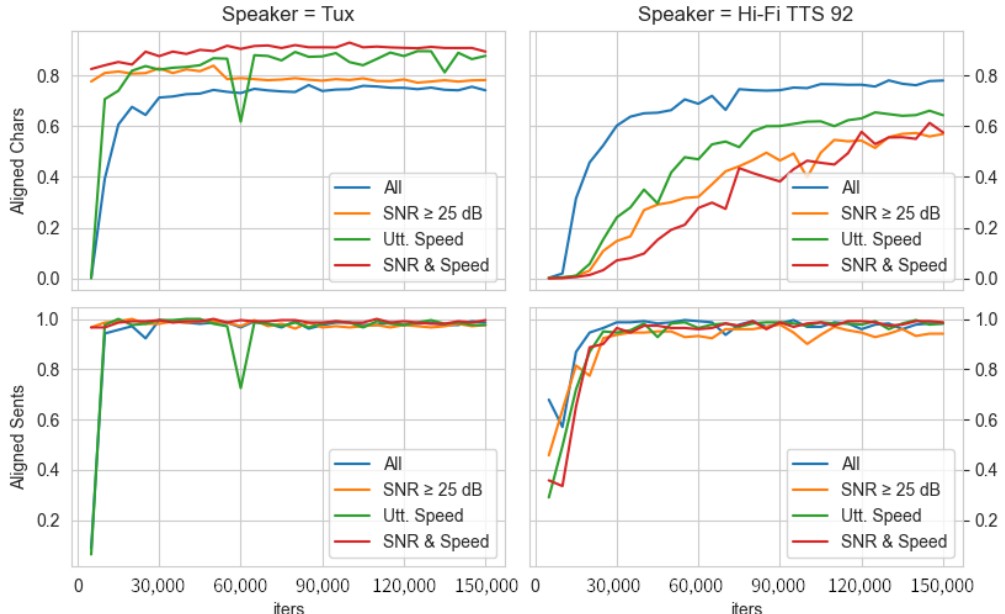

**Figure 9.** Fraction of aligned characters (**above**) and sentences (**below**) for the audio generated for each iteration for the HQ speakers *Tux* (**left**) and *Hi-Fi TTS 92* (**right**) trained from scratch on the whole corpora.

Concerning the fraction of aligned characters, however, two contrary trends can be witnessed for the two speakers. In the case of *Tux*, the performed ablations had a positive impact on the fraction of aligned input characters, even in the earlier stages of training. In relation to the speaker *Hi-Fi TTS 92*, however, the opposite holds true, since the removal of data that were considered of higher variability negatively affected this particular metric. This discrepancy may arise from the notable disparity in the training data between the two corpora, since speaker *Tux* possessed a considerably larger number of audio hours compared to *Hi-Fi TTS 92*. Consequently, it became more feasible for *Tux* than for *Hi-Fi TTS 92* to exclude more challenging audio files from the training process without compromising the quality of the final model, at least as this metric is concerned.

## 7. Conclusions

In this work, we extended the quality evaluation of a voice-cloning task from a difficult corpus from our previous project [3] and compared the results on a set of two higher-quality voice cloning corpora in both English and Spanish.

We first evaluated the quality of each corpora in terms of phonetic coverage, the SNR, and utterance speed. We performed an exhaustive evaluation of these features in order to detect which partitions of audio files could infer higher variability and, therefore, be detrimental to the quality of a voice-cloning model.

Using these data, a set of ablations was performed on the original datasets by removing audio files that were considered of lower quality—in terms of a low SNR—or higher variability—concerning utterance speed—from the training partitions. Audio files with an SNR lower than 20 dB for the more-difficult speaker and 25 dB for the higher-quality datasets were removed. Similarly, audio files whose utterance speed was in the first and last deciles were also withdrawn from the data. In addition and since the quantities of the three corpora were not fully comparable, these same ablations were applied to a randomly chosen 3 h subset of the two corpora featuring higher quality. In this regard, a total of 20 models were trained in order to check the impact of said audio removals.

In order to automatically check the quality of the trained models, we gathered a set of four different measurements. First, two different MOS estimators based on deep learning techniques were employed: NISQA [9] and MOSnet [4]. Additionally, we introduced a novel algorithm in order to complement the forced alignment of sentences. This approach takes advantage of the diagonal present in a successful synthesis in the attention matrix from the voice-cloning system Tacotron-2 in order to count the number of correctly aligned input characters.

With the aid of these measurements, we proved that removing data that were considered noisier or that featured a more-variable utterance speed from the more-difficult dataset improved the overall quality of the final models when starting from a pre-trained model, even though that half of the audio files were withdrawn from training in the harshest ablation. Moreover, the estimated MOS increased around 0.2 points for both algorithms.

Regarding the two datasets that featured a higher quality, since the level of noise present in these two corpora was comparatively low, the impact of only removing audio files with lower values of the SNR did not particularly help the training of the models. Nevertheless, the quality of the models with fewer data, but a more homogeneous utterance speed can be considered to be equal to those models with a higher amount of audio when using a pre-training approach in terms of the fraction of aligned characters.

Finally, when training from scratch using the whole dataset, the impact of removing more variable data was seen to be negative for the speaker with a fewer number of hours in terms of the fraction of aligned characters. Regarding the estimated MOS, however, the ablations did not show any significant deterioration of the MOS values estimated by NISQA and MOSnet, not in the pre-trained framework with 3 h of audio, nor when training from scratch with the whole dataset, which means that similar quality can obtained by using much fewer training data. Concerning this point, one of the main conclusions drawn from our work is that the quality obtained by training using 3 h of data from a pretrained model—even on a different language—or by training from scratch using more voluminous datasets was relatively similar, according to the objective metrics used. These observations suggested that considerably fewer data are needed for training a voice-cloning model if provided a robust pre-training as a starting point compared to training the model from scratch with a greater amount of data, with no substantial difference in the overall obtained quality. Therefore, this approach facilitated the process of corpus gathering for this specific task as fewer data were needed for obtaining a high-enough-quality voice-cloning model.

Regarding future work, these automatic evaluations can be complemented by means of a subjective evaluation based on the real MOS using human evaluators. Moreover, the acquisition of novel metrics that leverage prosodic information could complement the characterisation of heterogeneous data in order to automatically identify the subsets exhibiting

higher variability within a specific voice-cloning dataset, accompanied by corresponding ablations for assessing the impact of their heterogeneity in the final quality. Finally, these evaluations could be performed on different voice-cloning frameworks, featuring other TTS paradigms and extending it to voice-conversion approaches, in order to check whether these results can be extrapolated to the broad spectrum of architectures.

**Author Contributions:** Conceptualisation, A.G.-D. and A.Á.; methodology, A.G.-D. and A.Á.; software, A.G.-D.; validation, A.G.-D. and A.Á.; formal analysis, A.G.-D. and A.Á.; investigation, A.G.-D. and A.Á.; resources, A.G.-D. and A.Á.; data curation, A.G.-D.; writing—original draft preparation, A.G.-D.; writing—review and editing, A.G.-D. and A.Á.; visualisation, A.G.-D. and A.Á.; supervision, A.Á.; project administration, A.Á. All authors have read and agreed to the published version of the manuscript.

**Funding:** This work has been partially funded by the Spanish Public Business Entity Red.es, attached to the Ministry of Economic Affairs and Digital Transformation, under the C005/21-ED programme and in the context of the IANA project with reference 2021/C005/00152165.

**Institutional Review Board Statement:** Not applicable.

**Informed Consent Statement:** Not applicable.

**Data Availability Statement:** The two public corpora presented in this study are openly available at https://github.com/carlfm01/my-speech-datasets (accessed on 4 July 2023) and https://www.openslr.org/109/ [36] (accessed on 4 July 2023). Due to the particularities involving the personality of the corpus *XRey*, the data will not be shared.

**Conflicts of Interest:** The authors declare no conflict of interest.

## Abbreviations

The following abbreviations are used in this manuscript:

| | |
|---|---|
| CNN | Convolutional Neural Network |
| GAN | Generative Adversarial Network |
| GMM | Gaussian Mixture Model |
| HMM | Hidden Markov Model |
| HQ | High Quality; refers to speakers *Tux* and *Hi-Fi TTS* |
| MFCC | Mel Frequency Cepstral Coefficient |
| MOS | Mean Opinion Score |
| MFA | Montreal Forced Aligner |
| NISQA | Non-Intrusive Speech Quality Assessment |
| SNR | Signal-to-Noise Ratio |
| TTS | Text-To-Speech |
| VAD | Voice Activity Detection |

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
