# Peer review of "Enhancing Voice Cloning Quality through Data Selection and Alignment-Based Metrics"

_applsci, doi:10.3390/app13148049_

Round 1

Reviewer 1 Report

Thank you for your study. I have listed down my major and some minor concerns. 

1.     Please check the spelling and other grammatical mistakes present throughout the paper.

2.     Please explain all formulas in sentences. Indicate for what purpose the variables used are used.

3.     Please give the long version of the abbreviations where they first appear in the article.

4.     Please put your more important conclusions about the results in the summary.

5.     References started with reference number 2. This is not an acceptable situation. Please rearrange the references in order, starting with 1.

6.     The second paragraph started directly with, However. You should start this with a supporting sentence and give a reference.

7.     Please reference two distinct high-fidelity Voice Cloning datasets

8.     How exactly does this work differ from your other work? What are the similarities?

9.     What are the highlights of this work? Please indicate the contributions and highlights of this work at the end of the introduction.

10.  How was SNR measured?

11.  Please detail all sections 3.1-3.5.

12.  The algorithm description is insufficient; it should be detailed.

13.  Create a discussion section. Draw a table and compare it with other studies. Discuss the advantages and disadvantages of the study.

14.  Add more results and discuss them properly.

15.  Also mention some future research directions in the conclusion section.

Extensive editing of English language required

Author Response

Firstly, the authors would like to thank the reviewer for their interesting suggestions and contributions, which have led to a significant improvement of the article. Please, find the answers of the authors to each suggestion in the following points:

Point 1 of the reviewer: Please check the spelling and other grammatical mistakes present throughout the paper.

Response from the authors: Linking it with point 3, we modified the text in order to introduce the expanded versions of the acronyms the first time they appear in the text. Moreover, following the suggestion of another reviewer, we changed the plural of the word “audio” and substituted it by other forms such as “audio files” or “utterances”. We would be grateful to the reviewer if they had any other specific suggestions regarding the grammar or the spelling of our work.

Point 2 of the reviewer: Please explain all formulas in sentences. Indicate for what purpose the variables used are used.

Response from the authors: Linking it with point 12, we extended the explanation of the variables shown in the paragraph explaining the alignment algorithm in section 5.2.2 for further clarification.

Point 3 of the reviewer: Please give the long version of the abbreviations where they first appear in the article.

Response from the authors: We noted that some of the abbreviations were not indeed expanded the first time they appeared in the paper. We have checked the paper in order to fix these incoherences. For the case of the acronyms that refer to the name of a metric or a model (VITS, PESQ, POLQA or NISQA), we preferred to leave them as such instead of including their expanded version.

Point 4 of the reviewer: Please put your more important conclusions about the results in the summary.

Response from the authors: Linking it with point 9, the introduction section has been modified to include a paragraph with the conclusions obtained from this work. The reviewer may find it in the 9th paragraph of the Introduction:

“The difficulty of the dataset corresponding to the successful case presented in our previous work is due in part to the quality of the data, as well as to the lack of enough training data. However, we managed to prove that reducing the variability of this corpus by excluding specific subsets from the training partition improves the quality of the generated audio. In the case of the contrasting High Quality (HQ) datasets, the addition of more variable data does not necessarily imply an improvement of the synthesised audio when using the same training configuration than the difficult corpus, although the models trained from scratch using a higher amount of data were more notably influenced by these changes.”

Point 5 of the reviewer: References started with reference number 2. This is not an acceptable situation. Please rearrange the references in order, starting with 1.

Response from the authors: We removed the reference appearing in the note below the title so the references in the main text start with number 1.

Point 6 of the reviewer: The second paragraph started directly with, However. You should start this with a supporting sentence and give a reference.

Response from the authors: A reference of our previous work, that supports this evidence, was added to the text.

Point 7 of the reviewer: Please reference two distinct high-fidelity Voice Cloning datasets

Response from the authors: If we correctly understood the reviewer's suggestion, sections 3.2 and 3.3 include the references to both HQ corpora: for Hi-Fi TTS, the cite to the paper is included. In the case of Tux, since no publication is associated with this corpus, the link to the Github repository is linked on a footnote. We would be grateful if the reviewer informed us of any missing information they considered relevant so we can add it and improve this section of the work.

Point 8 of the reviewer: How exactly does this work differ from your other work? What are the similarities?

Response from the authors: Following the reviewer's suggestion, we added a more clarifying paragraph in the introduction linking our previous work with the presented work. The reviewer may find it in the 5th paragraph of the introduction:

“This work is an extension of the authors’ previous study, where we examined two real use cases of Voice Cloning system construction under highly challenging data conditions, characterised by a small quantity of highly variable and low-quality data. We analysed one successful and one unsuccessful case using objective metrics, without conducting exhaustive data curation. In this work, we expand upon the previous study by applying data selection techniques to the same successful case presented in the previous paper. Additionally, we introduce two new cases using two distinct high-fidelity Voice Cloning datasets publicly available in the community, one in English and the other in Spanish, and with varying corpus sizes. Furthermore, we use two objective evaluation metrics based on MOS estimation models and introduce a novel algorithm for calculating sentence alignment of synthesised audio at character level. Through this, we propose a methodology that concludes that data selection improves, or at the very least does not worsen, the quality of the voice cloning system used in this evaluation. Moreover, we propose an iterative evaluation of models during training using the aforementioned objective metrics, providing support for the construction of these systems before subjectively evaluating the final model.”

Point 9 of the reviewer: What are the highlights of this work? Please indicate the contributions and highlights of this work at the end of the introduction.

Response from the authors: Similarly to the previous point, another paragraph including the highlights of this work was added in the Introduction section, as indicated in the 4th point of the reviewer. The reviewer may find it in the 9th paragraph of the Introduction:

“The difficulty of the dataset corresponding to the successful case presented in our previous work is due in part to the quality of the data, as well as to the lack of enough training data. However, we managed to prove that reducing the variability of this corpus by excluding specific subsets from the training partition improves the quality of the generated audio. In the case of the contrasting High Quality (HQ) datasets, the addition of more variable data does not necessarily imply an improvement of the synthesised audio when using the same training configuration than the difficult corpus, although the models trained from scratch using a higher amount of data were more notably influenced by these changes.”

Point 10 of the reviewer: How was SNR measured?

Response from the authors: We extended the information regarding the measurement of the SNR in section 3.4.2: 

“In this study, the forced audio alignments generated in the previous step were used to obtain speech and non-speech segments required for SNR calculation, instead of relying on an external VAD module or WADA based algorithms. Using that information, the SNR values were computed by comparing the power of the speech vs the non-speech segments, supposing that the speech signal is the sum of the clean speech and the background noise obtained from the non-speech segment.”

Point 11 of the reviewer: Please detail all sections 3.1-3.5.

Response from the authors: We added more detail on the information of the corpora of section 3. For XRey, we summarised the data acquisition part from our previous work, including the sources and the alignment and post-editing sections. For Hi-Fi TTS and Tux, we extended the information regarding the original source of the audios. 

We also included information regarding the audio sampling and encoding of three corpora. 

We also extended information about the calculation of SNR as previously mentioned in the previous point.

Point 12 of the reviewer: The algorithm description is insufficient; it should be detailed.

Response from the authors: In addition to point 2, we extended the information on the variables in the paragraph that explains the algorithm for further clarification to the readers. Similarly, we substituted the logical and symbol (∧) for the word “and” in the pseudo-code.

Point 13 of the reviewer: Create a discussion section. Draw a table and compare it with other studies. Discuss the advantages and disadvantages of the study.

Response from the authors: As far as we know, there is no such study that measures the impact of reducing the most variable data from a highly heterogeneous corpus using objective metrics. We could compare the values of MOS obtained from both estimators NISQA and MOSnet with MOS values from other studies, but these comparisons do not offer a reliable set of metrics, since the comparison at MOS levels should take into account the variability between evaluators and should be performed on equal conditions. If the reviewer knew of the existence of a work that reproduces similar experiments, we would gratefully add these contributions to our work.

Point 14 of the reviewer: Add more results and discuss them properly.

Response from the authors: The results that were added in the paper are those considered the most significant ones in the whole study. We computed more intermediate results regarding the training of the different models, but we considered some of these intermediate results were not providing any more relevant information and worsened the clarity of the graphics provided. We also know that there are some other metrics involved in speech quality estimation, such as ViSQOL, but due to their characteristics we considered that they do not really fit in the nature of our work. However, if the reviewer has any other suggestions for improving the quality of our work regarding this issue, we would be glad to study them.

Point 15 of the reviewer: Also mention some future research directions in the conclusion section.

Response from the authors: In order to clarify future work for the readers, we rephrased the future research paragraph in the conclusion section in order to give future research directions to readers. It can be found at the end of the Conclusion section:

Regarding future work, these automatic evaluations can be complemented by means of a subjective evaluation based on real MOS using human evaluators. Moreover, the acquisition of novel metrics that leverage prosodic information could complement the characterisation of heterogeneous data in order to automatically identify the subsets exhibiting higher variability within a specific Voice Cloning dataset, accompanied by corresponding ablations for assessing the impact of their heterogeneity in the final quality. Finally, these evaluations could be performed on different Voice Cloning frameworks, featuring other TTS paradigms and extending it to Voice Conversion approaches, in order to check whether these results can be extrapolated to the broad spectrum of architectures.”

Reviewer 2 Report

In order to improving the quality of voice cloning, this manuscript investigate the impact of various techniques, two high-quality corpora were also employed for comparation purpose.

The manuscript proceeds in a straightforward, logical manner, and has been well written. I have three suggestions for authors to improve their nice work. 

1.      Because this is the extended work from author’s previous project (ref. 1), it would be better if authors could give more introduction about their previous work somewhere in the manuscript;

2.      The Acknowledgments section should be rewritten;

3.      Should the “Acoustics” in the “Copyright” section in the first page should be “Acoustics and Vibrations”, or “Applied Sciences”?

Author Response

Firstly, the authors would like to thank the reviewer for their interesting suggestions and contributions, which have led to a significant improvement of the article. Please, find the answers of the authors to each suggestion in the following points:

Point 1 of the reviewer: Because this is the extended work from author’s previous project (ref. 1), it would be better if authors could give more introduction about their previous work somewhere in the manuscript;

Response from the authors: As suggested, we included a further explanation and linking of our previous work in the introductory section of the journal. (Please, notice that we removed the cite from the title, so our previous project has now moved to the ref. 3) The reviewer may find it in the 5th paragraph of the introduction:

“This work is an extension of the authors’ previous study, where we examined two real use cases of Voice Cloning system construction under highly challenging data conditions, characterised by a small quantity of highly variable and low-quality data. We analysed one successful and one unsuccessful case using objective metrics, without conducting exhaustive data curation. In this work, we expand upon the previous study by applying data selection techniques to the same successful case presented in the previous paper. Additionally, we introduce two new cases using two distinct high-fidelity Voice Cloning datasets publicly available in the community, one in English and the other in Spanish, and with varying corpus sizes. Furthermore, we use two objective evaluation metrics based on MOS estimation models and introduce a novel algorithm for calculating sentence alignment of synthesised audio at character level. Through this, we propose a methodology that concludes that data selection improves, or at the very least does not worsen, the quality of the voice cloning system used in this evaluation. Moreover, we propose an iterative evaluation of models during training using the aforementioned objective metrics, providing support for the construction of these systems before subjectively evaluating the final model.”

Point 2 of the reviewer: The Acknowledgments section should be rewritten;

Response from the authors: We thank the reviewer for noticing the wrongly submitted acknowledgements section. We modified this section. 

Point 3 of the reviewer: Should the “Acoustics” in the “Copyright” section in the first page should be “Acoustics and Vibrations”, or “Applied Sciences”?

Response from the authors: We also thank the reviewer for noticing the wrong category of MDPI journal. We changed the category to Acoustics instead of Applied Sciences.

Reviewer 3 Report

In this work, the authors described their investigations on trimming the training datasets used for voice cloning neural networks, and showed that in some cases, removing lower quality samples from the dataset can improve the network performance, when evaluated using certain objective metrics.

Overall, the paper is fairly well written and presented. 

There are only a few grammar errors that could be fixed, such as the word "audios" should be "audio". The authors are recommended to proof-read before publishing.

Author Response

Firstly, the authors would like to thank the reviewer for their interesting suggestions and contributions, which have led to a significant improvement of the article. 

Point 1 of the reviewer: There are only a few grammar errors that could be fixed, such as the word "audios" should be "audio". The authors are recommended to proof-read before publishing.

Response from the authors: When using the plural form of "audios" we most of the time were referring to "audio files". Nevertheless, considering the comments of the reviewer and for the clarity of the reader, we fixed this particular issue and changed this plural version to "audio", "audio files" or "utterances" depending on the context.

Round 2

Reviewer 1 Report

Is superscript 3 in Table 3 used correctly?

The algorithm explanation is still insufficient. The algorithm should be explained in more detail.

The discussion part of the article is insufficient.

Please explain all formulas in sentences. Indicate for what purpose the variables used are used.

Figure 4 description is insufficient; it should be detailed.

The results section of the study should be expanded.

Extensive editing of English language required

Author Response

The authors thank the reviewer for their interesting suggestions and contributions, which have led to a significant improvement of the article. Please, find the answers of the authors to each suggestion in the following points:

Point 1 of the reviewer: Is superscript 3 in Table 3 used correctly?

Response from the authors: In order to avoid confusions with superscripts and numbers, the mathematical notation (1 < t < 10) has been replaced with letters (Short sentences).

Point 2 of the reviewer: The algorithm explanation is still insufficient. The algorithm should be explained in more detail.

Response from the authors: We modified the explanation of the algorithm in order to be clearer: 

“Let A = (aij) be an attention matrix of dimensions E × D where E is the length of the input sequence, D is the length of the output spectrogram, and every aij represents the attention score of the matrix. In order to check if all these elements have scored a minimum threshold value, they are clustered using sliding rectangles of a fixed size with values of width w and height h. The algorithm starts with the first rectangle at position x = 0, y = 0 (lower-left corner). Every element of the attention matrix aij inside said rectangle is checked to be higher than a given threshold value θ, which can be modified as an input parameter in order to be more or less aggressive. The rectangle is given a margin of ⅓w to the left in order to facilitate the search of aligned characters inside the region. If there exists a value aij > θ inside this rectangle, then the i-th input character is considered to be correctly aligned. This process is then repeated sliding the rectangle to the position of the last correctly aligned character until:

  • no correctly aligned character is found inside the region: there is no correctly synthesised character beyond this point,
  • its uppermost point exceeds E: the characters at the end of the sentence are aligned,
  • or its leftmost point exceeds D: the alignment algorithm reaches the end of the spectrogram.

We added comments and changed the names of some of the variables in order to make the pseudo-code clearer. Finally, we changed the caption of Figure 4 for further clarification of the algorithm, as shown in Point 5. 

Point 3 of the reviewer: The discussion part of the article is insufficient.

Response from the authors: We extending the final paragraph of the conclusions stating one of the main conclusions of the work:

Finally, when training from scratch using the whole datasets, the impact of removing more variable data is seen to be negative for the speaker with a less number of hours in terms of the fraction of aligned characters. Regarding the estimated MOS, however, the ablations did not show any significant deterioration of the MOS values estimated by NISQA and MOSnet, not in the pre-trained framework with 3 h of audio nor when training from scratch with the whole dataset, which means that similar quality can obtained by using much less training data. Concerning this point, one of the main conclusions drawn from our work is that the quality obtained by training using 3 h of data from a pretrained model – even on a different language – or by training from scratch using more voluminous datasets is relatively similar, according to the objective metrics used. These observations suggest that considerably less data is needed for training a Voice Cloning model if provided a robust pre-trained as a starting point compared to training the model from scratch with a greater amount of data, with no substantial difference in the overall obtained quality. Therefore, this approach facilitates the process of corpus gathering for this specific task as less data is needed for obtaining a high enough quality Voice Cloning model.

Point 4 of the reviewer: Please explain all formulas in sentences. Indicate for what purpose the variables used are used.

Response from the authors: We have changed the algorithm explanation paragraph present in Section 5.2.2, the pseudo-code shown in Algorithm 1, and the caption of Figure 4, including the renaming of some of the variables, in order to clarify the explanation. We also made sure that the link to the formula used for calculating the utterance speed (equation 1) is present on 2nd point of section 5.1, which explains the use of the name S. Finally, we removed the mathematical notation in Table 3, as explained in point 1. 

Point 5 of the reviewer: Figure 4 description is insufficient; it should be detailed.

Response from the authors: We extended the description of Figure 4. We also modified the figure in order to clarify the algorithmic procedure:

“An example of an attention matrix of a decoding in Tacotron-2 (left) and once highlighting the regions processed by the character alignment algorithm (right). Starting from the lower left corner, a character i is considered aligned if any value aij of said row i is higher than the threshold value θ = 0.7 inside a rectangle of width w = 150 and height h = 8 (marked in orange as rectangle 1). Then, the rectangle is slid up and right to the position of the lastly aligned character, following the diagonal, as indicated by the increasing numbers. May the reader notice that the size of the rectangle can be smaller if a region falls outside the attention matrix.”

Point 6 of the reviewer: The results section of the study should be expanded.

Response from the authors: We added a clarifying paragraph at the end of the Section 6.1 that draws the main conclusions of the experiments regarding the speaker XRey:

As the main conclusion to this case, in which low quality audios are involved, it can be stated that excluding the audios with the higher variability at SNR and speed levels help not only improve the quality of the models, but also reach better MOS estimator metrics faster.

The work was also modified in different sections for improving its readability, such as the order of the metrics in Section 6.1. 

Round 3

Reviewer 1 Report

The authors made the necessary corrections.

Extensive editing of English language required